# Measuring Community Greening Merging Multi-Source Geo-Data

**Weiying Gu [1,2], Yiyong Chen [1,3,*]  and Muye Dai [4]**

[1] Key Laboratory of Urban Land Resources Monitoring and Simulation, Ministry of Land and Resources, P.R.C., Shenzhen 518060, China; guweiy@163.com
[2] Pingshan Center for Urban Planning and Land Affairs of Shenzhen, Shenzhen 518118, China
[3] Shenzhen Key Laboratory of Built Environment Optimization, School of Architecture and Urban Planning, Shenzhen University, Shenzhen 518060, China
[4] High School Department, Shenzhen Experimental School, Shenzhen 518055, China; daixping@139.com
* Correspondence: chenyiy@szu.edu.cn

**Abstract:** Urban residential greening provides opportunities for social integration and physical exercise. These activities are beneficial to promoting citizens' mental health, relieving stress, and reducing obesity and violent crimes. However, how to measure the distribution and spatial difference of green resources in urban residential areas have been controversial. This study takes the greening of urban residential units in Shenzhen City as its research object, measures the various greening index values of each residential unit, and analyses the spatial distribution characteristics of residential greening, regional differences, and influencing factors. A large sample of street view pictures, urban land use and high-resolution remote sensing image data are employed to establish an urban residential greening database containing 14,196 residential units. This study proposes three greening indicators, namely, green coverage index, green view index, and accessible public green land index, for measuring the green coverage of residential units, the visible greening of surrounding street space and the public green land around, respectively. Results show that (1) the greening level of residential units in Shenzhen City is generally high, with the three indicators averaging 32.7%, 30.5%, and 15.1%, respectively; (2) the types of residential greening differ per area; and (3) the level of residential greening is affected by development intensity, location, elevation and residential type. Such findings can serve as a reference for improving the greening level of residential units. This study argues that one indicator alone cannot measure the greenness of a residential community. It proposes an accessible public green land index as a measure for the spatial relationship between residential units and green lands. It suggests that future green space planning should pay more attention to the spatial distribution of green land, and introduce quantitative indicators to ensure sufficient green lands around the walking range of residential areas.

**Keywords:** green coverage index; green view index; accessible public green land index; greening characteristics; residential units; residential greening

---

## 1. Introduction

Urban greening, such as woodlands, shrubs, grasslands, and other green spaces, is a widely recognized element of urban landscapes [1,2]. The ecological effects of urban greening include carbon sequestration, air purification, heat island effect alleviation and stormwater runoff filtration and reduction [3,4]. Greening in residential and surrounding areas is an important part of urban greening. Residential urban greening provides opportunities for social integration and physical exercise, which are beneficial to promoting citizens' mental health, relieving stress, and reducing obesity and violent

crimes [5,6]. Even though urban greening also has some positive impact on human health, excess pollen leads to respiratory allergies [7], yet it is undisputed that contact with nature and urban green space can have various positive impacts on human health and well-being.

In comparison with urban parks, street greening and neighborhood trees play more important roles in the lives of residents. They form green corridors in residential areas and streets, thereby providing multi-dimensional—but sometimes unnoticed—benefits [8–10]. For instance, street greening can reduce mental and physical stress and increase the perceived safety of urban streets [10]; street trees are associated with lowering asthma risk in children; green walking environments can prolong the life of the elderly; the presence of trees can affect people's behavior, such as encouraging children to walk to school and people to walk and cycle [9]. Research shows that about 90% of the information received by people in the environment comes from vision. The most commonly seen green landscape amongst residents is 3-D greening, as represented by street greening and neighborhood trees [11]. The measure of 3-D greening enables an intuitive understanding of the greening level in residential areas.

Extant green space evaluation and research mainly focus on 2-D urban greening [12] but rarely on street greening and neighborhood trees in residential areas [9]. For many years, China's planning management system evaluated urban greening on the basis of green rate, green coverage index, and park land per capita [13]. These indicators can evaluate intuitively and quantitatively the total amount of urban green space [14]. However, they cannot evaluate, guide, and control the distribution of greening in residential areas, and the relationship between green space and residential areas, especially the shortage of residential green space services amongst disadvantaged groups. These indicators can evaluate the 2-D greening effect but not the 3-D green resource distribution and landscape effect.

The concept of 'green view index' complements existing standards in the greening and visibility of street greening and neighborhood trees in residential areas [15]. Recognized by the international community, the green view index indicates the proportion of visible greens. Studies have shown that a green view index larger than 30% brings mental and psychological comfort to people, thereby inducing good physiological and psychological effects [15]. In comparison with parks and green spaces that serve as fixed greening centers, street trees, neighborhood trees and 3-D greening outside buildings are more flexible. They are the most visible green spaces amongst urban residents because they form green corridors along streets and pedestrian passages [16]. As an essential component of urban green space, street greening has substantially contributed to the attractiveness and walkability of streets [8]. The visibility of greening in residential areas helps increase the public's satisfaction with their living environment and considerably promote public health [17]. Few studies have systematically analyzed the distribution of urban residential areas and associated green spaces, including trees, lawns, neighborhood trees and other green spaces along the streets [18].

The traditional green view index data entail time-consuming and laborious acquisition and analysis [19]. Previous research on the green view index mainly used mechanical sampling and camera shooting to obtain photographs and then used Photoshop or other methods to estimate the green view index [20]. Data acquisition and processing consume much effort and resources, yet the amount of data is limited and the system error is large. Existing studies on the green view index generally have small sample sizes because they focus on few cities or parts of a city. Only a few works have systematically explored the spatial distribution differences and influencing factors of the green view index and its relationship with other indicators [18]. In recent years, the availability of data and the progress of computer image processing technology enabled in-depth research on the green view index. Additional channels to acquire urban spatial landscape images were created. Google, Tencent, Baidu and other street view map applications have been used for quantitative studies on the green view index. Street view photos can be obtained directly, efficiently, and easily from the internet, which guarantees large sample sizes [21]. Moreover, advancements in image processing technology lead researchers to come up with image processing algorithms and use MATLAB for quick batch computing of the green view index [22]; MATLAB can reduce errors caused by the manual identification of green blocks, thereby improving efficiency.

This study tries to emerge multiple indicators to measure urban residential greening of China. Currently, urban residential types in China are classified into commercial housing, collective dormitories, affordable housing, and urban villages of different densities. The supply of green space is also diversified: public green lands and street green spaces provided by the government, green spaces inside residential units provided by the developer, private gardens provided by individuals and balcony greening. Hence, the present study aims to answer the following question: Is there a spatial difference in the level of green resources in urban residential spaces? If yes, what are the influencing factors? How can we improve the greening level of urban residential space in the future? This study employs a large sample of street view images and other data to build a green database integrated by multi-source geographic data combined with existing urban greening indicators. It also explores methods for measuring greening level, spatial distribution, and regional differences amongst urban residential units.

## 2. Data and Methods

### 2.1. Case Selection

This study selected Shenzhen, one of the garden cities in China, as its research subject. Located in southern China, Shenzhen is one of the most developed cities in the country and is China's first special economic zone (SEZ). It is adjacent to Hong Kong and has a total area of 1952 km$^2$ and a total population of 12.53 million at the end of 2017 (Figure 1). Hence, it is the largest metropolis with the highest population density in China. As of 2017, its forest area reached 793.39 km$^2$ with a forest coverage rate of 40.04%; the green areas from gardens in the city's built-up areas covered 362.67 km$^2$, with a 39.2% green rate of the built-up area; the park land per capita was 16.04 m$^2$; and the green coverage index of the built-up area was 45.08% [23].

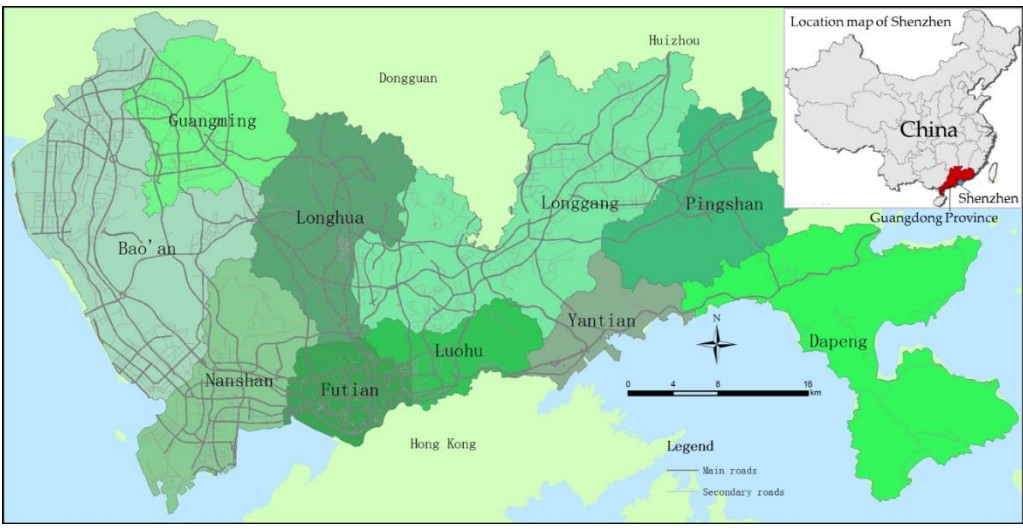

**Figure 1.** Location and administrative division map of Shenzhen.

Shenzhen City is a representative of China's rapid urbanization in the last 40 years. It has made considerable achievements in urban construction and ecological environment protection. It also leads the country in terms of park construction and greening of residential areas, as well as serves as a representative role in the urbanization of developing countries and regions [24]. As a young city, Shenzhen is also less constrained by the lack of green space apparent amongst old cities. Therefore, it is selected to explore the difference in green resource supply of urban residential space.

## 2.2. Residential Unit and Type Identification

This study used residential units as an analytical sample. A database on the current land use from the Shenzhen Land Planning Department contains 123,325 basic land units and indicates the nature of land use of each land parcel in 2014. This study used the residential parcels in the database as a sample unit. ArcGIS 10.2 spatial analysis tools was used to calculate the area of residential buildings on each parcel based on the Second Buildings Census of Shenzhen City (2014) and the land use in 2014. Approximately 97.1% of residential buildings were located on four land-use types: urban residential, rural residential, industrial, and military lands. The major property types on these land-use types are commercial housing (generally developed by a real estate development company approved by relevant government departments on state-owned land, and used for sale and rental of houses, including commercial residential buildings, affordable housing), informal housing (including urban villages and limited property rights houses that are developed on collective land), factory dormitories (dormitories located on industrial land) and military property houses (housed and real estate developed on military land), respectively. Altogether, these property types involve 19,575 land-use units. The remaining 2.9% of residential buildings are scattered on 36 land-use types, which involve 5359 land-use units. As such, the study focused on four major residential types and used the land parcels of the four types of land use as basic residential units. It also calculated the base and building areas of each building in each parcel. According to its land use and building type, 'apartment' was not identified as 'commercial housing'. The number of apartments in Shenzhen is small and their land-use types are very diverse. Commercial housing was divided into high-density (floor area ratio $\geq 2$) and low-density (floor area ratio $< 2$) commercial housing to further distinguish the greening characteristics of commercial houses at different densities (Table 1).

**Table 1.** Statistics of different residential units.

| Type of Residential Unit | Number of Residential Units | Gross Residential Area (m$^2$) |
|---|---|---|
| Informal housing | 3737 | 192,216,070 |
| High-density commercial housing | 4582 | 248,956,472 |
| Low-density commercial housing | 1202 | 36,327,790 |
| Military housing | 51 | 3,124,590 |
| Factory dormitory | 10,003 | 136,687,441 |
| **Total** | **19,575** | **617,312,363** |

## 2.3. Calculation of Three Greening Indicators

The study integrated existing methods to measure urban greening. It proposed three greening indicators, namely, green coverage index (hereinafter referred to as GCI), accessible public green land index (hereinafter referred to as GLI) and green view index (hereinafter referred to as GVI, Table 2).

GCI is employed to measure the green coverage of residential units. The scope of green coverage includes residential parks, greening between houses, roof greening, canopy of large trees, etc., focusing on the total amount of various green spaces in residential areas and the comprehensive ecological environment benefits. GLI is an index to measure public green land around the walking range, which focuses on the accessibility and recreation function of park lands. GLI value is sensitive to large green spaces, especially large mountain parks. GVI is employed to measure the visible greening of surrounding street space. As represented by street greening, neighborhood trees, three-dimensional greening, which are distributed everywhere, cheap and diverse. GVI focuses on the potential health effects of walking environments.

GVI was calculated using street view pictures. Studies have shown that regardless of pictures, virtual reality, or realistic green landscapes, regardless of whether they are aware of it or not, no significant difference was detected in the benefits that green landscapes bring to human beings [11]. Thus, we collect street view pictures from Tencent Street View via web crawler and location query (https://map.qq.com/). The capture angle was uniformly set to 0° (i.e., head up). The same street was

sampled at 50 m intervals when capturing images. Wide streets were sampled along both sides. The main and auxiliary roads were sampled separately, and intersections were regarded as the start and end points of sampling. Images of the front, back, left, and right directions were obtained for each sampling point. Each picture contained information such as the unique identifier of the point, latitude and longitude and orientation. We captured street view data of all streets in Shenzhen at June 2017. A total of 292,000 sample points were obtained with 1.168 million street view photos.

**Table 2.** A comparison of three greening indicators.

| Indicator | GCI (Green Coverage Index) | GLI (Green Land Index) | GVI (Green View Index) |
|---|---|---|---|
| Full spelling | Green coverage index | Accessible public green land index | Green view index |
| Definition | An index to measure the green coverage of residential units | An index to measure public green land around the walking range of residents | An index to measure the visible greening of surrounding street space of residential area |
| Objection | Total amount of various green spaces, including residential parks, greening between houses, roof greening, canopy of large trees, etc. | Various public green lands including urban, theme, community and country parks, open space, mountainous parks | All green elements on street view, including trees, shrub, lawn, vertical greening, even painted green wall |
| Data source | High-resolution satellite remote sensing image data | Citywide land-use vector data from the planning agency | Street view photos collected from tencent street view map |
| Calculation | The proportion of the green coverage area within the land area of the residential unit | The ratio of various public green lands within the residential unit and 500 m buffer zone of the residential unit, to the total area | The ratio of the number of green pixels to the total number of pixels in street view photos |

The automatic calculation of GVI has two steps: analysis of the colour composition of the street view images and aggregation of point-based GVI (Figure 2). The analysis of colour composition of the street view image was completed by the algorithm function in MATLAB. The colour model of each photo was converted from RGB to HSV. The values of the respective channels were extracted from the digital image. For each pixel, the degree (0–360°) of its colour in the HSV spectrum was calculated; 60–180° was defined as green. The GVI of each street view image was acquired through the ratio of the number of green pixels to the total number of pixels. The GVI of a point is the average GVI of four street view images at the point. The GVI of the residential unit is the average value GVI of all the sampling points inside the unit and the surrounding street space (a 100 m buffer zone outside the land boundary).

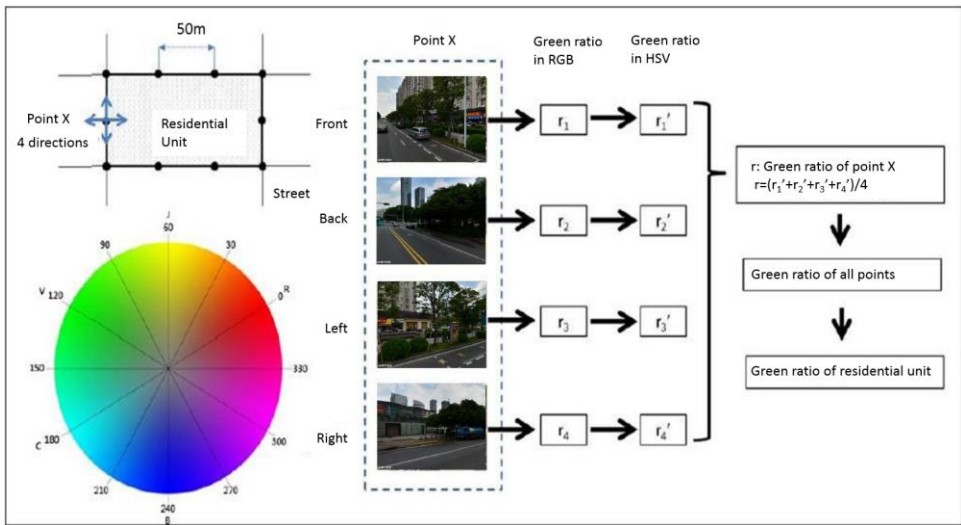

**Figure 2.** Calculation framework of the GVI, Image source: Long & Liu 2017.

GCI was calculated based on high-resolution satellite remote sensing image data. The remote sensing processing software was used to fuse the panchromatic and multi-spectral bands of satellite data. The spectral characteristics of multi-spectral bands were employed to extract urban green coverage information. The vector data of urban greening coverage were obtained after field verification and manual correction. This study used a multi-spectral, full-colour SPOT-5 image obtained on 30 November 2013. PCI Geomatica was used for fusion and ENVI was used for image processing to obtain images covering the whole city of Shenzhen. The object-oriented method was used for image segmentation, whilst the support vector machine was used for supervised classification to obtain the green coverage map of Shenzhen [25]. The GCI of the residential unit is the proportion of the green coverage area within the land area of the residential unit.

In terms of GLI, this study integrated accessibility and green rate to quantitatively compare the service level of public green land around different residential units [26]. We proposed an indicator of public green land index of a residential unit within walking distance (GLI). The calculation of GLI was based on current land-use data. The proportion of public green land within a certain distance around the residential unit was measured. According to the European Environmental Agency, people should be able to reach an urban green space within 15 minutes of walking, which is equivalent to 900–1000 m. The minimum green supply target set by the Netherlands is 60 m$^2$ per household within 500 m of the residential area, while the British standard suggests a minimum area of 2 ha within 300 m [26]. China's 'National Evaluation Standards of Garden Cities' proposed that 'the layout of urban public green land should be reasonable with a service radius of 500 m'. By contrast, 'Shenzhen City Planning Standards and Guidelines' proposed that the service radius of community parks should be 300–500 m. In the present study, a 500 m buffer zone of the residential unit was used. The GLI was calculated as follows: On the basis of the citywide land-use vector data, the ratio of various public green lands (including urban, theme, community and country parks, open space, mountainous parks, according to [27], where golf courses are not considered as they open to only a few people), within the residential unit and 500 m buffer zone of the residential unit, to the total area was calculated.

## 2.4. Database Establishment and Statistical Analysis

We established a comprehensive database of multi-dimensional greening of the 19,575 residential units identified by the above method, using the ArcGIS 10.2 spatial analysis tool. To minimize statistical error, residential units with less than 10 green-view sample points (occasional high or low values will affect the average of the green-view), or small parcels with areas less than 2000 m$^2$, or with total residential floor areas less than 1000 m$^2$ (considering the resolution of our remote sensing image, the small parcels and residential units have very high statistic error, which will affect the significance of the results) were excluded. The final number of residential units that were included in the statistical analysis was 14,196. The study measured three greening indicators of each residential unit: GCI, GVI and GLI. It analyzed the spatial distribution characteristics, regional differences and influencing factors of greening of residential units.

## 3. Results

### 3.1. Overall Greening Level and Correlation amongst Different Indicators

The GCI of residential areas in Shenzhen City is relatively high, with an average GCI inside residential units of 32.7% (Table 3). This value is lower than the GCI of the built-up area published in the relevant statistics (45.08%) [23]. The latter includes a large number of urban green land and forestry lands. The average value of GLI was 15.1%, which was slightly lower than the public green land rate (19.0%) of the city's built-up area. The average GVI was 30.5%, which was slightly higher than the 30% standard recommended by relevant literature [15].

Amongst the three greening indicators, GCI had the maximal average value, whereas GLI had the minimal average value. GLI and GCI had large standard deviations, whereas GVI had the

smallest. Accordingly, the amount of greening and public green land within different residential units differed greatly. However, little difference was observed in the visible greening landscape around the residential units.

**Table 3.** Basic attributes of residential units.

| Variables | Mean Value | Standard Deviation | Maximum Value | Minimum Value |
|---|---|---|---|---|
| Parcel area (m$^2$) | 24,786.75 | 34,286.35 | 787,620.81 | 2000.01 |
| Number of buildings in each unit | 35 | 84 | 3142 | 1 |
| Gross floor area (m$^2$) | 52,038.56 | 70,226.51 | 1,106,262.79 | 1216.80 |
| Gross residential area (m$^2$) | 36,714.03 | 60,994.06 | 990,345.75 | 1000.99 |
| Building density | 0.473 | 0.165 | 1.000 | 0.001 |
| Floor area ratio | 2.66 | 2.28 | 27.79 | 0.01 |
| GLI | 0.151 | 0.149 | 0.982 | 0.000 |
| GVI | 0.305 | 0.084 | 0.726 | 0.050 |
| GCI | 0.327 | 0.137 | 0.922 | 0.000 |

The high GCI inside residential units is related to the strict real estate development policy on green land in residential areas in China. The 'Code of Urban Residential Areas Planning & Design' stipulates that the green land rate should not be lower than 30% in new urban areas and not lower than 25% in rebuilt old areas. Commercial houses in Shenzhen are mostly located in new residential areas with a high green land rate. The GCI is even higher than the green rate because the land areas of tree projection, roof garden and grass-planting bricks are included in its calculation.

The GLI is 15.1% in average, and the average public green area in the 500 m buffer zone of residential units is 11.9 ha, which is higher than the 4.4 ha of a recent Western study [26]. The large number of mountain parks results in a relatively high public green land rate in Shenzhen. The 'Code for Classification of Urban Land Use and Planning Standards of Development Land (China)' proposes that the ratio of green space and urban square should be 10.0–15.0%. In addition, the surface area of mountains in Shenzhen City accounts for 62% of its total land area. Given that many residential units are built besides hills or mountains, residents can reach mountain or country parks within 500 m. However, a high standard deviation reflects a large spatial difference in the distribution of public green land.

The average GVI of residential units in Shenzhen is 30.5%, which indicates a high level of street greening level around residential units. However, the standard deviation of GVI is small, which reflects small differences in street greening around different residential units. Shenzhen's greening management regulations dictate that street greening is directly invested, constructed, and managed by the government. The municipal and district governments are responsible for the main and auxiliary roads. Therefore, the spatial difference of the greening level around residential units is very small.

A significant positive correlation was observed amongst the three indicators. The correlation coefficient between GVI and GCI was 0.241 (significance < 0.001); the correlation coefficient between GVI and GLI was 0.187 (significance < 0.001); and the correlation coefficient between GCI and GLI was 0.022 (significance < 0.01). The measurements of the greening level of the three indicators overlapped to some extent. They measure the greening inside a residential unit, the visible greening of surrounding street space and the public green land within walking distance, respectively. Therefore, the greening measured by GVI and that by the other two indicators overlapped, which explains its strong correlation. The measurement for GCI and GLI overlapped much less. Thus, their correlation is weak.

## 3.2. Spatial Differences of Residential Greening

The residential greening varies in different areas. On a citywide scale, the residential greening level in the east is higher than that in the west (Figure 3). The east area has more mountains and forests than the west, whereas the west has more plains and fewer forests. Moreover, residential greening in the south is better than that in the north because the former is more developed. In terms of differences

between districts inside and outside the original SEZ, both GVI and GCI of residential units in the original four SEZ districts (Luohu, Futian, Nanshan and Yantian) are higher than those in the six districts outside the original SEZ (Figure 4).

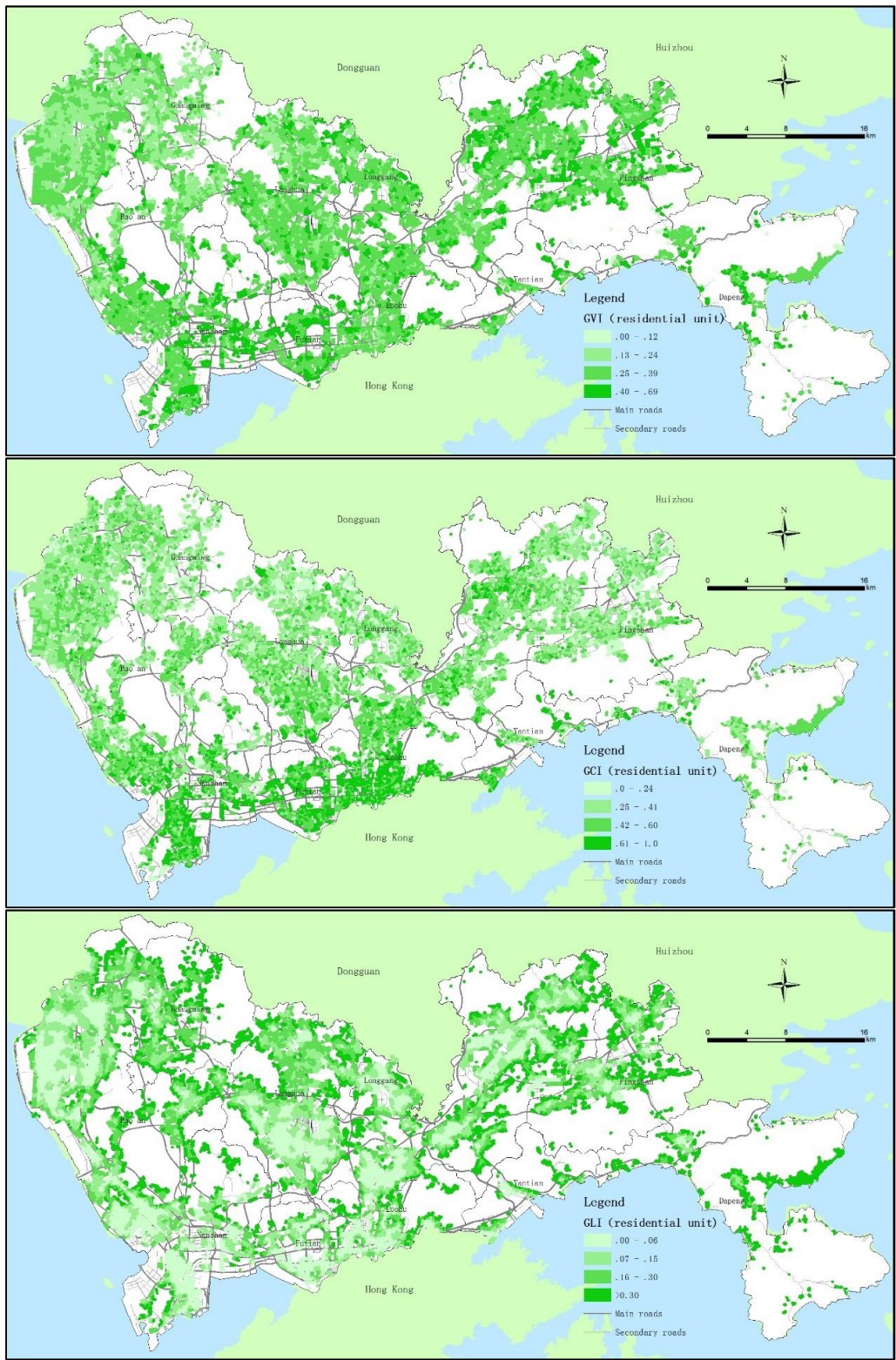

**Figure 3.** Distribution of three greening index values of residential units. GVI (top)/GCI (middle)/GLI (bottom).

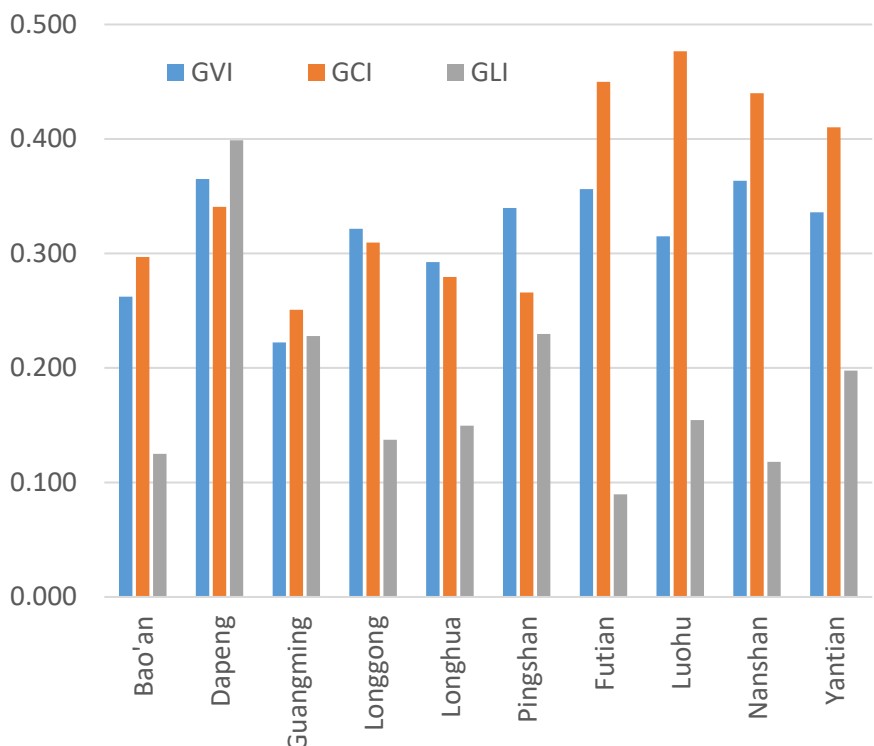

**Figure 4.** Mean values of three greening indicators in different districts.

In general, GCI value is much bigger than GLI, because the GLI measure focuses on public green land, except that in Dapeng district, which has a very high rate of public green land. In most districts, the value of GVI and GCI is almost the same, with some exceptions in the original four SEZ districts.

In terms of GCI, the housing market in the original four SEZ districts is relatively mature, thus their overall GCI is also relatively high, especially for commercial housing of all kinds, which has a 41–47% GCI. The GCI of all other districts outside the original SEZ is relatively low because their development started relatively late. Currently, these areas have many informal housing and industrial dormitories. Moreover, their building density is high, leaving few spaces for greening in residential areas. Hence, their average GCI is relatively low, between 25% and 34%.

In terms of GVI, each district differs slightly in terms of GVI. The average value of each district falls between 22% and 37%. Amongst the districts inside and outside the original SEZ, Dapeng and Pingshan district have relatively high greening levels. Dapeng district has the highest average GVI of 36.5%, followed by Nanshan District and Futian District at 36%. Guangming, Baoan and Longhua districts outside the original SEZ have a low average GVI value.

The average GLI of residential units in the whole city is 15.1%. Considerable differences were observed amongst districts whose average value falls between 9% and 40%. Dapeng District still has the highest value, with a GLI of 39.9%, followed by Pingshan, Guangming and Yantian with GLI of above 20%. These areas have many forests and country parks, and many residential areas are built on the mountainside. Futian, Nanshan and Baoan districts have the lowest GLI. Generally, the number and area of the mountain parks in these areas are small. Their public green lands are mainly comprehensive parks and small mountain parks. Hence, they have low GLI value.

*3.3. Greening Differences amongst Different Types of Residential Units*

Different types of residential units present large greening differences (Figure 5). Low-density commercial and military housings have the highest greening levels, with high GVI, GCI and GLI value. By contrast, informal housing and factory dormitories have the lowest greening levels, with lower GVI and GCI value. Military housing comprises real estate developments that rely on military facility land.

Most houses in such developments are located in good environments and have high greening levels. High-density commercial housing has the lowest GLI. However, low-density commercial housing has the highest GLI because many of them (including villas) are geographically located in well-greened areas, such as places around country parks and mountain basins. Commercial houses built in the early stage of urban development are mostly low density. Most of them are located in places with a good natural environment, around a comprehensive park, for example.

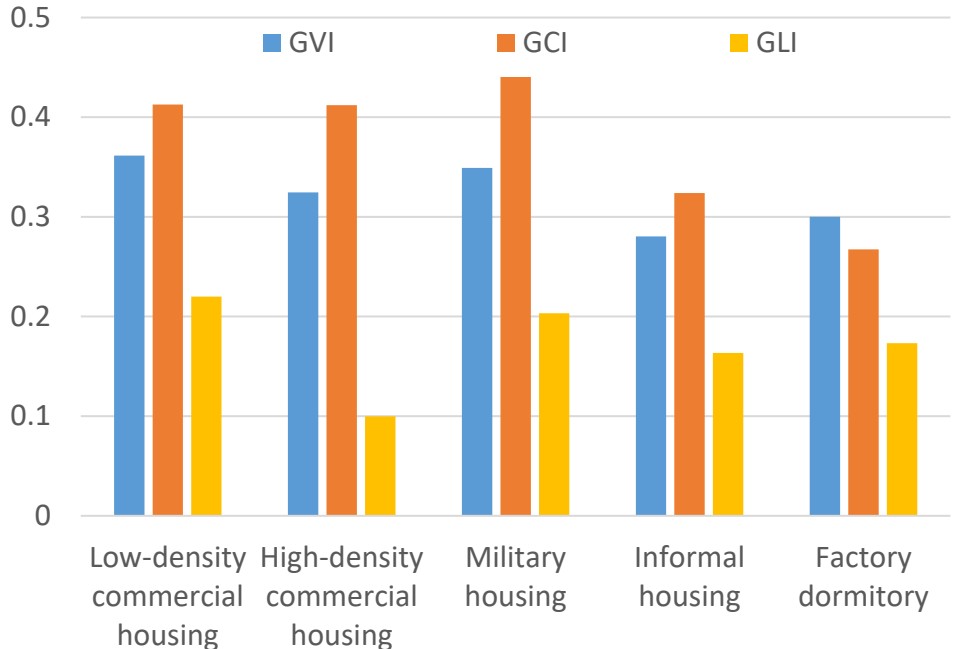

**Figure 5.** Mean values of three greening indicators for different residential types.

Informal housing and factory dormitories have the lowest greening levels. Their GVI are low because most of these residential spaces are located on collective village land, where streets are generally narrow, with less or no greening. Their GCI is also very low. The building densities of these residential spaces are generally very high. As most of these spaces are built spontaneously by villagers or workers, they lack unified planning management, which leaves limited space for greening. These residential spaces also have fewer public green lands within walking distance, reflecting the overall low greening investment by the village collective.

*3.4. Factors Affecting Residential Greening*

To explore the factors affecting the greening of residential units, we selected 12 variables and established linear regression models in an attempt to identify the geospatial factors related to the residential greening level. Considering that the dependent variables are a continuous variable and of normal distribution, the study uses linear regression models. Detail of the model refers to several previous studies [3,28,29]. The variables include the characteristics of the residential unit itself (i.e., parcel area, number of buildings, building density, floor area ratio, whether it is a commercial house or a factory dormitory), location of the residential unit (i.e., whether it is in the original SEZ, distance from the CBD, the number of bus lines around) and the surrounding environment of the residential unit (i.e., whether it is backed by a mountain, whether it is facing water, elevation). When selecting the variables, we consulted related research [3,21] and considered the availability of data and common knowledge.

Three linear regression models were established using the Enter method in SPSS 22.0. The dependent variables are GVI, GCI and GLI, respectively. The VIF values of all variables in the models are less than three. Hence, no significant collinearity exists amongst the variables. The corrected

regression coefficients for each variable are 0.195, 0.319 and 0.265, respectively. Thus, the models could explain some of the factors associated with residential greening level, especially GCI model.

Amongst the three models, the symbols (positive and negative correlations) of most variables are consistent (Table 4). Therefore, these variables have a consistent influence on the three greening indicators. The regression coefficients of building density, floor area ratio and distance from CBD are negative (and significance < 0.001), whereas those of elevation—whether it is inside the original SEZ and whether it is commercial housing—are positive (and significance < 0.001). Thus, residential units with high greening levels have the following characteristics: low development intensity (low building density and small floor area ratio), close to CBD, high elevation, located in the original SEZ or belong to commercial housing. Amongst the other factors, the number of buildings (negative correlation) and whether it is a factory dormitory (negative correlation) significantly affect the GVI. By contrast, the GCI is significantly affected by whether the residential unit is a factory dormitory (negative correlation) and its parcel area (negative correlation). Specifically, the larger the number of buildings inside a residential unit, the lower the GVI; and the larger the parcel area, the lower the GCI.

**Table 4.** Standardized coefficients and significance level of the regression model.

| Variable Types | Variables | GVI | GCI | GLI |
|---|---|---|---|---|
| Characteristics of residential units | Parcel area | 0.018 | −0.053 ** | −0.019 |
| | Number of buildings | −0.073 ** | −0.030 * | −0.016 |
| | Building density | −0.179 ** | −0.151 ** | −0.106 ** |
| | Floor area ratio | −0.043 ** | −0.071 ** | −0.051 ** |
| | Whether it is commercial housing (yes = 1) | 0.172 ** | 0.101 ** | 0.050 ** |
| | Whether it is a factory dormitory (yes = 1) | −0.088 ** | −0.154 ** | −0.026 |
| Location | Whether it is in the original SEZ (yes = 1) | 0.363 ** | 0.273 ** | 0.312 ** |
| | Distance from CBD | −0.195 ** | −0.067 ** | −0.324 ** |
| | Number of bus lines | −0.016 | 0.018 | −0.038 |
| Surrounding environment | Whether it is backed by a mountain (yes = 1) | 0.032 | 0.011 | −0.006 |
| | Whether it is facing water (yes = 1) | −0.061 * | −0.008 | −0.021 * |
| | Elevation | 0.274 ** | 0.053 ** | 0.260 ** |
| | Sample quantity | 14196 | 14196 | 14196 |
| | adj $R^2$ | 0.195 | 0.319 | 0.265 |
| | F value | 287.7 | 553.3 | 426.1 |

** Correlation is significant at the 0.001 level (two-sided). * Correlation is significant at the 0.01 level (two-sided).

## 4. Discussions and Conclusions

This study established a large sample urban residential greening database based on multi-source geographic data in Shenzhen City. It integrated existing methods for measuring urban greening and proposed three greening indicators, namely, GCI, GVI, and GLI, to measure the green coverage of residential units, the visible greening of surrounding street space and the proportion of public green land within the walking range, respectively.

Results show that residential greening differs greatly in different areas. The level of residential greening in the east is higher than that in the west. Moreover, the greening level in the districts inside the original SEZ is higher than that in the districts outside the original SEZ. The greening of different types of residential units also varies. The greening level of low-density commercial housing is the highest, whereas that of informal housing and factory dormitories is the lowest. Regression analysis finds that residential units with low development intensity (low building density and small floor area ratio), close to CBD, at high elevation, located in the original SEZ and belong to commercial housing have high greening levels, a finding that is basically consistent with the findings of related research [21]. To improve the level of residential greening, the present study suggests paying attention to the greening of residential units that are far from the CBD, at low elevation and located outside the original SEZ, as well as those that have high development intensity or are informal housing and factory dormitories.

This study shows that the three greening indicators have both commonalities and differences. The average GCI is the largest, whereas the average GLI is the smallest. Therefore, the greening measurements of the three indicators have overlaps, which explain the strong correlations amongst them. The greening level of residential units in Shenzhen City is generally high, with the averages of GCI, GVI, and GLI being 32.7%, 30.5% and 15.1%, respectively. The values of the three indicators, which are calculated based on large sample units, are consistent with the findings of the relevant statistical data and research [21,23]. The methods and indicators proposed in this study have practical significance and can be used to supplement the existing greening indicators of residential areas.

In addition, this study finds that the difference in spatial distribution of GLI is the largest amongst the several types of greening of residential units, whereas the spatial difference of GVI is the smallest. Accessible public green land (GLI) refers to urban parks, community parks, country parks, mountain parks and other urban public green lands. Relevant planning standards mainly control the total amount of urban public green land but fail to effectively control its spatial distribution, especially its relationship with residential units [30]. Street greening is invested, constructed and maintained, generally, by government departments at all levels, and thus the spatial difference of GVI is small. Considering the smallest spatial differences of GVI of residential units, small investments and ubiquitous health benefits, we recommend adding GVI indicators to measure differences in residential greening to better promote health and equity. In terms of GLI, due to the stipulation of residential greening in the 'Code of Urban Residential Areas Planning & Design' that 'green rate should not be lower than 30% in new urban areas and should not be lower than 25% in rebuilt old areas', the overall spatial difference is not large.

There has been numerous variety between green space indicators. Green land ratio and green coverage ratio are two popular indicators [31]. Previous studies mostly used rthe official land use dataset or data derived from aerial photography to measure the accessibility of public green land [32,33], or used public green land per capita as an indicator to measure the supply of public green land per capita [33,34]. However, both indicators failed to measure the spatial relationship between certain residential units and surrounding green spaces [35]. We integrated accessibility and green land rate for the first time in literature to our best knowledge. We proposed and measured the GLI and found that this indicator could well measure the spatial relationship between residential units and surrounding green spaces. In addition, it found that the spatial difference of GLI is greater than those of the GVI and GCI. Therefore, future green space planning should pay attention to the spatial distribution of public green land. Quantitative indicators should also be introduced to ensure the proportion of public green lands within walking distance of residential areas.

According to the correlation analysis, this study also found that the three indicators correlated, but with weak correlations. As stated in Table 2, three indicators measured the residential greenery from different aspects with different data sources. This finding suggests that researchers interested in using greenery measures cannot rely on one indicator alone to measure greenness of residential community. We recommend that, both accessible public green land, green coverage, and green view ratio indicators, should be added to the residential greening standards. And we suggest a minimum value for GLI, GVI, GCI, is 15.1%, 30.5%, 32.7% respectively, for future residential community development, according to the average value of our case.

**Author Contributions:** W.G. and M.D. participated the data processing and analysis. W.G. and Y.C. conceived the methodology and wrote the paper.

**Funding:** This research was funded by the Open Fund of Key Laboratory of Urban Land Resources Monitoring and Simulation, Ministry of Land and Resources.

**Acknowledgments:** We would like to thank CitoryTech for providing us technical support in this study.

**Conflicts of Interest:** The authors declare no conflict of interest.

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
