# Peer review of "Measuring Community Greening Merging Multi-Source Geo-Data"

_sustainability, doi:10.3390/su11041104_

Round 1

Reviewer 1 Report

- it's necessary to add the City of Shenzhen in the title

- there is some idealization of the health positive impact of the greening of urban space: greening the urban space often leads to worsening respiratory allergies. 

- it lacks a true integrated territorial typology of greening in Shenzhen's urban space

- there is also a lack of reference to golf courses, so important in the green spaces of Shenzhen.

- the question of the representativeness of the Shenzhen case was not asked

Author Response

- it's necessary to add the City of Shenzhen in the title

Response: we thank the reviewer for this important suggestion.

- there is some idealization of the health positive impact of the greening of urban space: greening the urban space often leads to worsening respiratory allergies. 

Response: we agree that the greening of urban space have positive impacts. We add relative description in introduction section.

- it lacks a true integrated territorial typology of greening in Shenzhen's urban space;- there is also a lack of reference to golf courses, so important in the green spaces of Shenzhen.

Response: we thank the reviewer for these excellent suggestions. We add some descriptions in section 2.3.

- the question of the representativeness of the Shenzhen case was not asked

Response: we thank the reviewer for pointing out this important issue. We have made necessary revisions in section 2.1.

Reviewer 2 Report

The paper entitled "How green are residential areas? An analysis of community greening emerging multi-source geo-data" studies the green level of urban residential areas in Shenzhen city (China). The author compares the overall greening level, spatial distribution and explanatory factors of three green indicators obtained using an imagery source, an official land-use dataset and street view photos. This shows some key differences between the later (the green view index encounters small green objects such as street trees which are managed by municipal and district governments and is equally distributed among the city) and the two other indices (which are mainly influenced by mountains, public parks and private lawns). This result proves that the interpretation of intra-urban spatial variations is affected by changes in the data source. This is for me a major contribution to the (massive) literature on urban green spaces. Besides, the paper reads very well and the method is described with great attention. Therefore, I believe that the paper should be published in your journal after minor improvements.

·         A discussion on the differences of health, environmental and recreational benefits get from the type of greens encompassed in each of the three indicators is missing. This is a pity as you show that their distribution differs between residential types. Also, you do not discuss the potential pressure on green spaces and how it may alter the benefits people get from them in densely populated areas. The fact that the GVI is better spread among the city could be explained by the fact that it is cheaper to plant trees than to build parks. It would be interesting if you could discuss the political and planning explanations of these disparities.

·         I would have also appreciated if you would have positioned more cautiously your paper’s contribution to the literature on urban green measurement (I’m thinking of articles such as the ones by Le Texier, Schiel and Caruso 2018 ; Van De Voorde 2016 ; Mitchell R, Astell-Burt T, Richardson 2011 for e.g.).

·         As two of your indicators show important standard deviations, I wonder whether the linear regression is the best statistical model to apply. Could you please describe the preliminary analyses that led you choosing linear regression models? It may be useful to add scatter plots.

·         The residential types are very specific to the Chinese case, could you please provide some descriptions? Besides, I am not sure of what you mean by “commercial housing”. Please be more specific.

·         In section 2.4, you explain that small parcels have been removed from your study. Please discuss how this may affect your results.

Less importantly:

-          Figure 2: I would increase the size of the legend and depict the blue spaces (sea) as well.

-          Figures 3 and 4 would be easier to interpret if you provide a map of the city districts.

-          Please change the main title of Figure 3 to “Mean values of three greening indicators in different districts” (“regions” is misleading).

-          Section 3.4, please be cautious with the interpretation of your R² : at the best, the model explain a third of the green index variations.

Author Response

The paper entitled "How green are residential areas? An analysis of community greening emerging multi-source geo-data" studies the green level of urban residential areas in Shenzhen city (China). The author compares the overall greening level, spatial distribution and explanatory factors of three green indicators obtained using an imagery source, an official land-use dataset and street view photos. This shows some key differences between the later (the green view index encounters small green objects such as street trees which are managed by municipal and district governments and is equally distributed among the city) and the two other indices (which are mainly influenced by mountains, public parks and private lawns). This result proves that the interpretation of intra-urban spatial variations is affected by changes in the data source. This is for me a major contribution to the (massive) literature on urban green spaces. Besides, the paper reads very well and the method is described with great attention. Therefore, I believe that the paper should be published in your journal after minor improvements.

Response: we thank the anonymous reviewer very much for reading the manuscript so carefully, and offering so many important and constructive comments. We highly appreciate the very helpful suggestions concerning the revision of the manuscript.

·         A discussion on the differences of health, environmental and recreational benefits get from the type of greens encompassed in each of the three indicators is missing. This is a pity as you show that their distribution differs between residential types. Also, you do not discuss the potential pressure on green spaces and how it may alter the benefits people get from them in densely populated areas. The fact that the GVI is better spread among the city could be explained by the fact that it is cheaper to plant trees than to build parks. It would be interesting if you could discuss the political and planning explanations of these disparities.

Response: we thank the reviewer for this constructive suggestion on a comparison of the three indicators. We add detail comparison in section 2.3 and discussion section, line 392-396.

·         I would have also appreciated if you would have positioned more cautiously your paper’s contribution to the literature on urban green measurement (I’m thinking of articles such as the ones by Le Texier, Schiel and Caruso 2018 ; Van De Voorde 2016 ; Mitchell R, Astell-Burt T, Richardson 2011 for e.g.).

Response: we thank the reviewer for this excellent suggestion. We made revisions in line 400-406.

·         As two of your indicators show important standard deviations, I wonder whether the linear regression is the best statistical model to apply. Could you please describe the preliminary analyses that led you choosing linear regression models? It may be useful to add scatter plots.

Response: we thank the reviewer for this important suggestion. We added relative description in section 3.4.

·         The residential types are very specific to the Chinese case, could you please provide some descriptions? Besides, I am not sure of what you mean by “commercial housing”. Please be more specific.

Response: we thank the reviewer for these important comments. We added detail description in line 133-139.

·         In section 2.4, you explain that small parcels have been removed from your study. Please discuss how this may affect your results.

Response: we thank the reviewer for this important reminding. We added relative discussion in section 2.4.

Less importantly:

-          Figure 2: I would increase the size of the legend and depict the blue spaces (sea) as well.

-          Figures 3 and 4 would be easier to interpret if you provide a map of the city districts.

-          Please change the main title of Figure 3 to “Mean values of three greening indicators in different districts” (“regions” is misleading).

-          Section 3.4, please be cautious with the interpretation of your R² : at the best, the model explain a third of the green index variations.

Response: we thank the reviewer for these important comments. We redraw figure 2 (revised as figure 3), add a map of the city district (figure 1) and changed the title of figure3. Also, we revised the statement in section 3.4.

Submission Date

16 December 2018

Date of this review

17 Jan 2019 11:50:06

Reviewer 3 Report

I have organised my comments and suggestions by section.  The only major comments are listed under results, with one in the methods section: how do you know your analysis didn't capture a green painted building and label it a tree?  A visual inspection of places where the GCI and GVI are highly negatively correlated (big residuals) might help you find out.  If you've already done some kind of cross check and have confidence this isn't a problem, it should be stated in the paper somewhere. Put another way: the description of the GVI needs more detail generally.

Introduction/Literature Review:

-Some of the introduction is very repetitive, the authors mention the benefits of greening twice over two paragraphs.  Perhaps spend the second paragraph elaborating on the benefits in a little more detail as they relate to your new measure of greening-e.g. your measure captures the greening that does the most good. 

-It's not clear to me why the authors use the terms "2-D" and "3-D" to describe a measure of parkland versus a measure of residential greenery--parks exist in three dimensions, do they not?  This term needs to be defined before it is used--my understanding is the '2D' and '3D' descriptors have more to do with the methods?

Methods:

-The paragraph justifying Shenzhen is persuasive but there are no citations to back up claims such as "It also leads the country in terms of park construction and greening of residential areas." It would be good to have a cite for this.

- I had to read the section very carefully to understand the actual construction of each measure.  I would bring those explanations to the top of each paragraph describing them, and then discuss/justify, at least for GCI and GLI.  You talk about different GLI standards and then in a brief sentence explain your measure.  I'd suggest starting with defining your measure in the paragraph and then referencing GLI standards that informed that approach.  You could also consider tossing the definitions/methods for each measure (plus data used) into a table.  If you are interested in seeing this method replicated elsewhere, that table could also include some measure or descriptor of the computing requirements.  GVI sounds very intensive. 

-Regarding the GVI: What about buildings/cars painted green?   What "greens" counted as green? How about flowers that are purple?  I'm sorry to nitpick here, but a little more detail in this section would be valuable as this is your main methodological contribution.

Results/Conclusions:

- I think the conclusions need to link back to the introduction.  Yes, you found that the three metrics correlated.  But correlations of .24, .18, .02 etc. are not very big.  What were the R^2s?  If only 60% of the GVI is explained by the GLI, for example, that finding suggests that researchers interested in using greenery measures cannot rely on GLI alone to measure greenness of a community.  I think that's a valuable finding. I could be wrong, I don't know, as you didn't explain the correlations of the three metrics in much detail.

-Related to the above comment, I think the comparison of the measures would be more interesting if they were standardized in some way, like percentile rank within the sample.  It might make the comparison more fruitful or revealing and make Figures 3 & 4 easier to digest. 

-Mapping the residuals of the correlations between the three metrics may offer more original insights than the current maps of the three figures. I imagine that, to researchers working in the Shenzhen context, the map findings are not that surprising, while maps of where the measures do not correlate could be far more interesting.

-Figures 3 and 4 are currently arranged to invite the reader to compare each measure across space and land use type, whereas I think the paper is more about comparing the three metrics--so the authors should consider re-arranging the figures to cluster results by district or land use, so we can see how each measure tells a different sore about that district or land use.

Author Response

I have organised my comments and suggestions by section.  The only major comments are listed under results, with one in the methods section: how do you know your analysis didn't capture a green painted building and label it a tree?  A visual inspection of places where the GCI and GVI are highly negatively correlated (big residuals) might help you find out.  If you've already done some kind of cross check and have confidence this isn't a problem, it should be stated in the paper somewhere. Put another way: the description of the GVI needs more detail generally.

Response: we thank the anonymous reviewer very much for reading the manuscript so carefully, and offering so many important and constructive comments. We highly appreciate the very helpful suggestions concerning the revision of the manuscript. They are not only important for the revision of the paper, but also very valuable for our future researches.

Specifically, we thank the reviewer for the important comments regarding GVI. Actually, we don't care whether GVI describes green trees or other greens. Some walls are painted green, and some are bare land temporarily covered in green, but the purpose of these “fake” greens is to increase the green color of the city. Studies have shown that regardless of pictures, virtual reality, or realistic green landscapes, even awareness or not, no significant difference was detected in the benefits that green landscapes bring to human being. We added relative revision in section 2.3, lines 160-170.

Introduction/Literature Review:

-Some of the introduction is very repetitive, the authors mention the benefits of greening twice over two paragraphs.  Perhaps spend the second paragraph elaborating on the benefits in a little more detail as they relate to your new measure of greening-e.g. your measure captures the greening that does the most good. 

Response: we thank the reviewer for this important suggestion. We revised the introduction section follow the reviewer’s suggestion.

-It's not clear to me why the authors use the terms "2-D" and "3-D" to describe a measure of parkland versus a measure of residential greenery--parks exist in three dimensions, do they not?  This term needs to be defined before it is used--my understanding is the '2D' and '3D' descriptors have more to do with the methods?

Response: We thank the reviewers for raising this question and helping us to articulate these two concepts more clearly. We made revisions in section introduction and 2.3.

Methods:

-The paragraph justifying Shenzhen is persuasive but there are no citations to back up claims such as "It also leads the country in terms of park construction and greening of residential areas." It would be good to have a cite for this.

Response: we thank the reviewer for this suggestion. We add a cite and some description in section 2.1.

- I had to read the section very carefully to understand the actual construction of each measure.  I would bring those explanations to the top of each paragraph describing them, and then discuss/justify, at least for GCI and GLI.  You talk about different GLI standards and then in a brief sentence explain your measure.  I'd suggest starting with defining your measure in the paragraph and then referencing GLI standards that informed that approach.  You could also consider tossing the definitions/methods for each measure (plus data used) into a table.  If you are interested in seeing this method replicated elsewhere, that table could also include some measure or descriptor of the computing requirements.  GVI sounds very intensive. 

Response: we thank the reviewer very much for presenting these important questions and suggestions. We reorganized section 2.3 and add a table to compare the three greening indicators.

-Regarding the GVI: What about buildings/cars painted green?   What "greens" counted as green? How about flowers that are purple?  I'm sorry to nitpick here, but a little more detail in this section would be valuable as this is your main methodological contribution.

Response: we thank the reviewer for pointing out this question and we have made some revisions on the manuscript in section 2.3, lines 167-170.

Results/Conclusions:

- I think the conclusions need to link back to the introduction.  Yes, you found that the three metrics correlated.  But correlations of .24, .18, .02 etc. are not very big.  What were the R^2s?  If only 60% of the GVI is explained by the GLI, for example, that finding suggests that researchers interested in using greenery measures cannot rely on GLI alone to measure greenness of a community.  I think that's a valuable finding. I could be wrong, I don't know, as you didn't explain the correlations of the three metrics in much detail.

-Related to the above comment, I think the comparison of the measures would be more interesting if they were standardized in some way, like percentile rank within the sample.  It might make the comparison more fruitful or revealing and make Figures 3 & 4 easier to digest. 

Response: we agree with the reviewer for these constructive suggestion on the finding of the manuscript. We made revisions in the last three paragraphs of the discuss section.

-Mapping the residuals of the correlations between the three metrics may offer more original insights than the current maps of the three figures. I imagine that, to researchers working in the Shenzhen context, the map findings are not that surprising, while maps of where the measures do not correlate could be far more interesting.

Response: we thank the reviewer for this inspiring suggestion. We tried this new method and mapped the residual of three regression analysis. However, the spatial differences are complicated and difficult to interpret to our best knowledge. So we intend to carry out such analysis in future research regarding the metrics.

-Figures 3 and 4 are currently arranged to invite the reader to compare each measure across space and land use type, whereas I think the paper is more about comparing the three metrics--so the authors should consider re-arranging the figures to cluster results by district or land use, so we can see how each measure tells a different sore about that district or land use.

Response: we thank the reviewer very much for this important suggestion. We re-arranged the figures and added relative analysis to interpret the new discoveries in section 3.2 and 3.3.

Submission Date

16 December 2018

Date of this review

10 Jan 2019 16:37:15